# Characterization of Three Novel Viruses from the Families *Nyamiviridae, Orthomyxoviridae*, and *Peribunyaviridae*, Isolated from Dead Birds Collected during West Nile Virus Surveillance in Harris County, Texas

**DOI:** 10.3390/v11100927

**Published:** 2019-10-10

**Authors:** Peter J. Walker, Robert B. Tesh, Hilda Guzman, Vsevolod L. Popov, Amelia P.A. Travassos da Rosa, Martin Reyna, Marcio R.T. Nunes, William Marciel de Souza, Maria A. Contreras-Gutierrez, Sandro Patroca, Jeremy Vela, Vence Salvato, Rudy Bueno, Steven G. Widen, Thomas G. Wood, Nikos Vasilakis

**Affiliations:** 1School of Biological Sciences, The University of Queensland, St Lucia QLD 4072, Australia; peter.walker@uq.edu.au; 2Department of Pathology, University of Texas Medical Branch, 301 University Blvd, Galveston, TX 77555, USA; rtesh@utmb.edu (R.B.T.); rbtesh@comcast.net (H.G.); vpopov@utmb.edu (V.L.P.); 3Center for Biodefense and Emerging Infectious Diseases, University of Texas Medical Branch, 301 University Blvd, Galveston, TX 77555, USA; 4Center for Tropical Diseases, University of Texas Medical Branch, 301 University Blvd, Galveston, TX 77555, USA; 5Institute for Human Infection and Immunity, University of Texas Medical Branch, 301 University Blvd, Galveston, TX 77555, USA; 6World Reference Center for Emerging Viruses and Arboviruses, University of Texas Medical Branch, 301 University Blvd, Galveston, TX 77555, USA; 7Mosquito and Vector Control Division, Harris County Public Health and Environmental Services, 3330 Old Spanish Trail, Houston, TX 77021, USA; nareyma@yahoo.com (M.R.); Jeremy.Vela@phs.hctx.net (J.V.); vence.salvato@hcpid.org (V.S.); rudybuenojr@gmail.com (R.B.); 8Center for Technological Innovation, Evandro Chagas Institute, Ananindeua, Para 67030-000, Brazilwmarciel@hotmail.com (W.M.d.S.); spatroca@gmail.com (S.P.); 9Virology Research Center, Ribeirao Preto School of Medicine, University of Sao Paulo, Ribeirao Preto, Sao Paulo 14025-000, Brazil; 10Program for Study and Control of Tropical Diseases (PECET), University of Antioquia and National University of Colombia, Medellin, Colombia; 11Department of Biochemistry and Molecular Biology, University of Texas Medical Branch, 301 University Blvd, Galveston, TX 77555, USA; sgwiden@utmb.edu (S.G.W.); TexRex47@comcast.net (T.G.W.)

**Keywords:** arboviruses, *Nyamiviridae*, *Orthomyxoviridae*, *Peribunyaviridae*, avian viruses

## Abstract

This report describes and characterizes three novel RNA viruses isolated from dead birds collected during West Nile virus surveillance in Harris County, TX, USA (the Houston metropolitan area). The novel viruses are identified as members of the families *Nyamaviridae*, *Orthomyxoviridae*, and *Peribunyaviridae* and have been designated as San Jacinto virus, Mason Creek virus, and Buffalo Bayou virus, respectively. Their potential public health and/or veterinary importance are still unknown.

## 1. Introduction

In 2002, when West Nile virus (WNV) was first detected in Texas [1], the Harris County Public Health Mosquito and Vector Control Division (HCMVCD), in collaboration with the University of Texas Medical Branch (UTMB) in Galveston, initiated a surveillance program to monitor WNV activity in wild birds in Harris County and the City of Houston. The avian surveillance program has been described previously [1,2,3]. Basically, it consisted of netting, bleeding, and releasing live wild birds and testing their sera for antibodies to WNV and St. Louis encephalitis virus (SLEV). Dead birds collected in the city and county were also tested for the presence of WNV [1,2].

From 2002 to 2014, a total of 9047 dead birds were submitted to UTMB for culture; 1644 of the birds (18.17%) yielded WNV. Other known viruses recovered from the brain cultures of dead birds from Harris County during this 13-year period included Newcastle disease virus (NDV), eastern equine encephalitis virus (EEEV), Mermet virus (MERV), and Flanders virus (FLAV) [4,5]. In this report, we describe and characterize three additional novel viruses from the families *Nyamaviridae*, *Orthomyxoviridae*, and *Peribunyaviridae* that were also isolated from dead birds in Harris County during the study period.

## 2. Materials and Methods

### 2.1. Study Area

Harris County, which includes the city of Houston, is located in the northern coastal plain of the Gulf of Mexico, a 50-m swath along the Texas Gulf Coast. Harris County covers a geographic area of more than 1768 square miles. It is the largest county in Texas; its human population was estimated to be 4,538,000 in 2015. The climate is classified as humid subtropical. The average annual rainfall in Houston is 1145 mm, with an annual mean temperature of 20.5 °C. A more complete description of the study area was given previously [1].

### 2.2. Collection of Dead Birds

As part of the WNV surveillance program, HCMVCD staff have collected dead birds found in public places or at private residences in the county since 2002 [1,2]. Bird carcasses were identified, tagged with location and date, placed in double plastic bags and stored in ‒70 °C freezers for subsequent transport on dry ice to the World Reference Center for Emerging Viruses and Arboviruses (WRCEVA) in the Department of Pathology at UTMB for testing.

### 2.3. Culture Methods

After thawing, the skull was opened with sterile surgical instruments and a small piece of brain was removed from each bird in a biosafety level 3 (BSL-3) laboratory. Brain tissue was homogenized using a TissueLyser (Qiagen, Hilde, Germany) in tubes with 1.5–2.0 mL of phosphate-buffered saline, pH 7.4, containing 10% fetal bovine serum, 1% penicillin‒streptomycin‒amphotericin stock (Sigma, St. Louis, MO, USA) and several 3-mm stainless-steel balls. After centrifugation, the supernatant was passed through a 0.20-µm nylon syringe filter (Fisher Scientific, Pittsburgh, PA, USA) to remove bacteria and fungi. Then 150 µL of the filtrate was inoculated into separate 12.5-cm^2^ flask cultures of Vero E6 and *Aedes albopictus* C6/36 cells, originally obtained from the American Type Culture Collection (Manassas, VA, USA). After adsorption for 2 h at 28 °C (C6/36) or 37 °C (Vero), 5.0 mL of maintenance medium was added to each flask, and they were held in incubators at 28 °C and 37 °C, respectively. Cell cultures were examined regularly for evidence of viral cytopathic effect (CPE). Two brain cultures, designated TX-9285 (DO 200) and TX-8339 (DO 159), produced CPE in C6/36 cell and Vero cell cultures, respectively, and yielded the three viruses described in this report.

### 2.4. Evaluation of Pathogenicity in Suckling Mice

Two litters of two-day-old ICR mouse pups (*n* = 20) were inoculated intracranially with approximately 15 µL of fluid medium from each of the two cell cultures showing CPE. After inoculation, the pups were returned to their dams and examined daily for 14 days for signs of illness or death. The mice were purchased from Harlan Sprague Dawley (Indianapolis, IN, USA); animal work at the University of Texas Medical Branch was carried out under an Institutional Animal Care and Use Committee-approved protocol (no. 9505045; approval date: May 1, 2013)).

### 2.5. Transmission Electron Microscopy (TEM)

For ultrastructural analysis, infected cells were fixed for at least 1 h in a mixture of 2.5% formaldehyde prepared from paraformaldehyde powder and 0.1% glutaraldehyde in 0.05 M cacodylate buffer (pH 7.3), to which 0.03% picric acid and 0.03% CaCl_2_ were added. The monolayers were washed in 0.1 M cacodylate buffer, and cells were scraped off and processed further as a pellet. The pellets were post-fixed in 1% OsO_4_ in 0.1 M cacodylate buffer (pH 7.3) for 1 h, washed with distilled water, and stained in a block with 2% aqueous uranyl acetate for 20 min at 60 °C. The pellets were dehydrated in ethanol, processed through propylene oxide, and embedded in Poly/Bed 812 (Polysciences, Warrington, PA, USA). Ultrathin sections were cut on a Leica EM UC7 µL tramicrotome (Leica Microsystems, Buffalo Grove, IL, USA), stained with lead citrate, and examined with a Phillips 201 transmission electron microscope (FEI Phillips, Hillsboro, OR, USA) at 60 kV.

### 2.6. RNA Extraction, Viral Genome Sequencing, and Assembly

Viral RNAs were extracted using TRIzol LS reagent (Invitrogen, Carlsbad, CA) as described previously [6]. Viral RNA (~0.9 µg) was fragmented by incubation at 94 °C for 8 min in 19.5 µL of fragmentation buffer (Illumina 15016648). A sequencing library was prepared from the sample RNA using an Illumina TruSeq RNA v2 kit following the manufacturer’s protocol. The samples were sequenced on a HiSeq 1500 using the 2 x 50 paired-end protocol. Reads in fastq format were quality-filtered, and any adapter sequences were removed using Trimmomatic software [7]. The de novo assembly program ABySS [8] was used to assemble the reads into contigs, using several different sets of reads from 25,000 to 1 million read pairs, and kmer values from 20 to 40. The longest contigs were selected from the results, the reads were mapped back to the selected contigs using bowtie2 [9], and finally visualized with the Integrated Genomics Viewer [10] to verify that the assembled contigs were correct. Sequence annotation was determined using the Geneious version 9.1.2 (Biomatters, Auckland, New Zealand) [11]. Sequences were deposited in the GenBank database (http://www.ncbi.nlm.nih.gov/genbank/) under the following access numbers for the isolates SJCV (MK971153), MCRV [MK037473 (PB1), MK037472 (PB2), MK MK037474 (PA), MK037475 (NP), MK037476 (HA) and MN450148 (M)], and BBAV [MK037470 (L), MK037471 (S)]. Raw sequencing data are available upon request to the corresponding author.

### 2.7. Viral Genome Annotation

The size, organization, terminal regions, open reading frames (ORFs), encoded proteins, and their conserved motifs were annotated using applications available in the Geneious suite. Protein functions were verified using InterPro [12]. The identification of transmembrane regions and signal peptides was performed in the TOPCONS webserver [13] and the identification of glycosylation sites in the NetNglyc 1.0 Server (http://www.dbs.dtu.dk/services/NetNGlyc/). The genetic relatedness of the isolates SJCV and MCRV, as well as the recovered genomes of BBAV and other viruses were first inspected by the Blastx algorithm available from the NCBI (http://www.ncbi.nlm.nih.gov/genbank/).

### 2.8. Phylogenetic Analyses

Multiple amino acid sequence alignments were conducted using ClustalX [14] or in MUSCLE [15] in MEGA version 7.0. Maximum likelihood phylogenetic trees were inferred in MEGA using the GTR (general time reversible) nucleotide substitution model and a discrete gamma distribution to model evolutionary rate differences among sites rates of substitution among sites. All positions containing gaps were eliminated. Initial trees for the heuristic search were obtained automatically by applying the neighbor-joining and BioNJ algorithms to a matrix of pairwise distances estimates using the maximum composite likelihood (MCL) approach and then selecting the topology with the superior log likelihood value. The phylogenetic robustness of each node was determined using 100 bootstrap replicates and nearest-neighbor branch-swapping. Trees were annotated in MEGA and final mark-up of trees was conducted in Adobe Illustrator CC (Adobe Inc., San Jose, CA, USA).

### 2.9. Serological Tests

Suckling mouse brain, sucrose‒acetone-extracted antigens, and mouse hyperimmune ascitic fluids (HIAFs) were prepared for isolate DO 200, as well as for Nyamanini (NYMV), Midway (MIDWV), Sierra Nevada (SNVV), and Newcastle disease (NDV) viruses, as described previously [16,17]. The NDV strain used was a 2005 Harris County isolate from a Eurasian collared dove (GenBank EU477193) [4]. Complement fixation (CF) tests were performed by a microtiter technique [18], using two units of guinea pig complement and overnight incubation of the antigen and antibody at 4 °C. CF titers were recorded as the highest dilutions giving CF levels of 3+ or 4+ on a scale of 0 to 4+.

## 3. Results

### 3.1. Virus Isolation and Initial Characterization

Sample DO 200 was the brain homogenate from a dead European starling (*Sturnus vulgaris*) collected in Crosby, Harris County, TX on 20 June 2013 (Table 1). The sample produced CPE in monolayer cultures of Vero E6 cells within 5–6 days after inoculation, but no CPE was observed following the inoculation of C6/36 cell cultures. Culture fluid from the infected Vero cells produced illness in newborn mice 4–5 days after intracranial inoculation.

Sample DO 159 was from a partially decomposed carcass of a blue jay (*Cyanocitta cristata*) collected in Katy, Harris County on 27 July 2011 (Table 1). During dissection, unidentified fly larvae (maggots) were observed in the brain. The homogenized filtered sample produced CPE (large syncytia) in the C6/36 cell culture from six days post-inoculation. No CPE was observed in Vero cells, and inoculation of culture fluid from the infected C6/36 cells did not produce illness after intracranial inoculation of newborn mice.

### 3.2. Transmission Electron Microscopy

In ultrathin sections of C6/36 cells infected with sample DO 159, numerous highly pleomorphic virus particles (spherical, filamentous, and of irregular shape) were observed to be forming at the cell surface (Figure 1a). Spherical particles were 130‒150 nm and up to 270 nm in diameter (Figure 1b); filamentous particles were 65–85 nm in diameter and could be very long, reaching 1–1.3 µm (Figure 1c). Ultrathin sections of Vero E6 cells infected with sample DO 200 revealed large virus particles, either spherical, from 320 nm to 800 nm in diameter, or pleomorphic and elongated, with sizes ranging from 350 × 540 nm to 280 × 1120 nm and 3280 × 1275 nm (Figure 1d,e). Virus particles formed at the cell surface in large quantities.

### 3.3. Nucleotide Sequence Analysis

Illumina sequencing of brain sample DO 200 yielded a single viral contig of 13,295 nt representing the genome of a novel virus named San Jacinto virus (SJCV). Of 12.8 million total read pairs, 961,704 (~7.5%) mapped to the SJCV sequence. A BlastX search of the against NCBI nonredundant protein sequence (nr) database indicated highest homology with the genomes of Midway virus (MIDWV), Nyamanini virus (NYAV), and Sierra Nevada virus (SNVV), each of which is an unsegmented, negative sense (-) single-stranded (ss) RNA virus assigned to the genus *Nyavirus*, family *Nyamiviridae*, order *Mononegavirales* [16,17,19]. The SJCV genome sequence appeared to be complete, except for the extreme 3’ and 5’ terminal regions. The genome contains seven long open reading frames (ORFs), each bounded by relatively conserved consensus sequences corresponding to transcription initiation (3’-CGUUGG-5’) and transcription termination/polyadenylation (3’-AGAAAUCUUUUU-5’) signals (Figure 2a). ClustalX alignments indicated that six of the ORFs encode proteins corresponding to the six recognized nyavirus proteins: ORF1 encodes the nucleoprotein (N) of 402 amino acids (aa) (45 kDa); ORF2 encodes the negative regulator of the viral polymerase (X) of 187 aa (estimated 21.4 kDa); ORF3 encodes the polymerase cofactor (P) of 373 aa (41.7kDa); ORF4 encodes the matrix protein (M) of 192 aa (22.2 kDa); ORF5 encodes the class I transmembrane glycoprotein (G) of 678 aa (76.5 kDa, pre-modification), which contains four predicted glycosylation sites (triangles Figure 2a); and ORF7 encodes the full-length RNA-dependent RNA polymerase (L) of 1943 aa (218.5 kDa). ORF6, which does not occur in the other nyaviruses, encodes a unique protein (designated P6) of 337 aa (38.9 kDa), which displays obvious amino acid sequence identity with the N-terminal domain of the SJCV L protein (23.1%), as well as the L proteins of other nyaviruses (19.8–24.9%) (Appendix A). The extent of sequence conservation indicates that ORF6 has most likely arisen through a partial gene duplication event. The duplicated region precedes the six conserved regions (CRI‒CRVI) containing polymerase, polyribonucleotidyl transferase, and methyl transferase domains that are present in the L proteins of all unsegmented (-) ssRNA viruses and its function has not been clearly defined. Amino acid sequence identity (p-distance) comparisons with the L proteins of other nyamiviruses confirmed that SJCV shares a close relationship (62.4–82.4% identity) with MIDWV, NYAV, and SNVV (Appendix A).

Illumina sequencing of brain sample DO 159 revealed sequences that BlastX searches indicated represented the genomes of two distinct viruses. Of 12.5 million read pairs, 147,260 (~1.2%) mapped to the genome of a virus that was named Mason Creek virus (MCRV) and 291,205 (~2.3%) mapped to the genome of a virus that was named Buffalo Bayou virus (BBAV). The MCRV genome was represented as six contigs that ranged in size from 968 nt to 2485 nt (Figure 2b). Each contained complete coding sequences that aligned with highest homology to six (-) ssRNA segments (PA, PB1, PB2, NP, HA, and M) of Longchuan virus (LCHV), which was isolated from mosquitoes (*Culex quinquefasciatus*) collected in Guangdong Province, China (https://www.ncbi.nlm.nih.gov/Taxonomy/Browser/wwwtax.cgi?id = 2594109), as well as Quaranfil virus (QRFV; genus *Quaranjavirus*, family *Orthomyxoviridae*, order *Articulavirales)* and several other currently unclassified viruses from ticks or birds that have been identified as possible quaranjaviruses [20,21]. The absence of conserved sequences with partial inverse complementary indicated that the 3’ and 5’ terminal noncoding regions were incomplete. The MCRV polymerase complex is formed by three segments: PA encoding a 773-aa protein (88.3 kDa); PB1 encoding a putative protein of 782 aa (90.6 kDa); and PB2 encoding a putative protein of 797 aa (91.8 kDa). The NP segment encodes the nucleoprotein of 523 aa (59.6 kDa). The HA segment encodes the hemagglutinin protein of 514 aa (58.5 kDa); the hemagglutinin is predicted to possess a cleavable signal peptide and three potential N-glycosylation sites. The M segment encodes the matrix protein of 281 aa (32.8 kDa). An exhaustive search of the available contigs failed to detect segment 7 of Wellfleet Bay virus (WFBV), which encodes a putative nonstructural protein VP7 [21], which we used as a surrogate sequence in our attempts to detect the seventh fragment of MRCV. Clustal X alignments of the deduced amino acid sequences of the PA, PB1, PB2, HA, and NP proteins indicated that MCRV was most closely related to LCHV, and more distantly related to QRFV, WFBV, Araguari virus (ARAV), Tjuloc virus (also named Tyulek virus; TLKV) [22], and Beihai orthomyxo-like virus 1 (BhOMV1) [23] (Appendix A).

The BBAV genome consisted of two (-) ssRNA segments of 1080 nucleotides (S RNA) and 7052 nucleotides (L RNA), each with very long terminal regions (5′ and 3′ NCR) of 219–886 nt and 96–138 nt, respectively (Figure 2c). The absence of conserved sequences with partial inverse complementary indicated that the 3’ and 5’ terminal noncoding regions were incomplete. BlastX searches indicated that the segments aligned with the genome segments of various members of the family *Peribunyaviridae*. The BBAV L segment encodes for the RNA-dependent RNA polymerase (RdRp) of 2272 amino acids with a predicted molecular weight of 264.5 kDa. The BBAV S segment encodes a nucleoprotein of 221 amino acids and predicted molecular weight of 25.6 kDa, but there was no evidence of an alternative open reading frame encoding a small nonstructural protein (NSs), as occurs in some members of the family *Peribunyaviridae* [24]. Exhaustive searches of available contigs also failed to detect sequences corresponding to the M segments of other peribunyaviruses that encode the two envelope glycoproteins (Gn and Gc). The BBAV RdRp shares a relatively low level of amino acid sequence identity (p-distance) with those of other peribunyaviruses (29.1–31.4%) (Appendix A), but contained all recognized functional domains including pre-motif A and motifs A through E (aa positions between 970 to 1240). The BBAV nucleoprotein is similar in size but shares a relatively low sequence identity (16.8–23.6%) with the other peribunyaviruses (Appendix A). Bunya-like viruses lacking an M segment have been reported previously in viral metagenomic studies of invertebrates [23,25]. However, the failure to detect this segment in BBAV and these other viruses may well be a consequence of the divergent nature of the sequences rather than their genuine absence in the viral genomes. 

### 3.4. Complement Fixation Tests

To determine their immunological relationships, CF tests were performed using mouse brain antigens and mouse HIAFs prepared to the SJCV, NYMV, MIDWV, SNVV, and NDV viruses (Table 2). SJCV MIAF had a homologous antibody titer of 1:128, but did not cross-react with the other antigens. Likewise, MIAF to the other viruses did not cross-react with SJCV.

### 3.5. Phylogenetic Analyses

Phylogenetic trees inferred using amino acid sequences for SJCV, MCRV and BBAV grouped the viruses in the families *Nyamiviridae*, *Orthomyxoviridae* and *Bunyaviridae*, respectively. Phylogenetic analysis of the RdRp (L protein) of SJCV and other members of the order *Mononegavirales* indicated that SJCV clusters with members of the genus *Nyavirus*, family *Nyamiviridae* (BSP = 100%) (Figure 3). The genus *Nyavirus* currently consists of three members: Midway virus (MIDWV; species *Midway nyavirus*), Nyamanini virus (NYMV; species *Nyamanini nyavirus*), and Sierra Nevada virus (SNVV; species *Sierra Nevada nyavirus*) [19]. The sequence data suggest that SJCV should be assigned to a fourth species in the genus *Nyavirus*. All four of these viruses have been associated with birds and/or ticks [16,17]. 

Phylogenetic analysis of the amino sequences of the MCRV RdRp (PB1, PB2, and PA) and nucleoprotein (NP) indicated that it clusters with members of the family *Orthomyxoviridae* (Figure 4a–d). The *Orthomyxoviridae* currently comprises seven genera (*Quaranjavirus*, *Thogotovirus*, *Alphainfluenzavirus*, *Betainfluenzavirus*, *Gammainfluenzavirus*, *Deltainfluenzavirus*, and *Isfavirus*) (https://talk.ictvonline.org/files/master-species-lists/m/msl/8266). Although the members of three of these genera infect birds, MCRV did not fall within any of the existing genera but clustered with high support (BSP >90%) with LCHV (from mosquitoes), as well as Beihai orthomyxo-like virus 1 (from woodlice), Jiujie fly virus (from horseflies), and members (assigned and proposed) of the genus *Quaranjavirus* [23,25]. The ecological diversity of these viruses suggests that MCRV should be assigned to a new species and genus within the *Orthomyxoviridae*.

Phylogenetic analysis of the amino acid sequences of the BBAV RdRp (L protein) indicated that it is a member of the family *Peribunyaviridae* [26]. BBAV lies within a well-supported clade that also includes members of the genus *Herbevirus* (Herbert virus [HEBV; species *Herbert herbevirus*], Kibale virus [KIBV; species *Kibale herbevirus*], Tai virus [TAIV; species *Tai herbevirus*]) [27], Shuangao insect virus 1 (SgIV-1; species *Insect shangavirus*) [25], and two currently unassigned viruses (Khurdun virus [KHUV] and Akhtuba virus [AKHV]) [28,29] (Figure 5). HEBV, KIBV, and TAIV were each isolated from mosquitoes in Africa and are thought to be insect-specific viruses as they replicate in insect cells but not in vertebrate cells [27]. SgIV-1 was detected by next-generation sequencing of a mixed insect pool containing lacewings (Chrysopidae) and moth flies (Psychoda) in China [23]. KHUV and AKHV were each isolated from birds in Russia, but do not cluster directly with BBAV. The data indicate that BBAV cannot be assigned at this time to any existing genus within the *Peribunyaviridae* [26] and so appears to represent a unique new species. Moreover, more genetic data are needed to accurately classify BBAV. Lastly, both MCRV and BBAV were isolated from the decomposing brain of a dead bird, which contained fly larvae (maggots). Based on their phenotypic characteristics (growth only in mosquito cells) and their phylogenetic relationship, they would appear to be insect-specific viruses, probably infecting the fly larvae and not the bird.

## 4. Discussion

The potential public health or veterinary importance of SJCV, MCRV, and BBAV is unknown, as are their prevalence and host range in Harris County. These are subjects for future investigation. Likewise, the pathogenesis of the three viruses should be examined in several avian and possibly mammalian species.

The discovery of the three novel viruses during our WNV surveillance program also reinforces the value of virus culture in arbovirus surveillance programs. Many public health and veterinary diagnostic laboratories now simply screen animal carcasses or pools of hematophagous insects by PCR or other nonculture techniques for the detection of RNA of viruses of interest (i.e., West Nile virus, Saint Louis encephalitis virus, eastern equine encephalitis virus, Venezuelan encephalitis virus, epizootic hemorrhagic disease virus, vesicular stomatitis virus, etc.). While these techniques are very sensitive, provide faster results, are more economical, and reduce the risk of accidental infection of laboratory personnel, they generally detect only the viruses for which one has appropriate primers. These techniques usually do not detect novel or unexpected viruses.

For discovering novel viruses, metagenomic analysis is now the current gold standard, as it allows the detection and genetic characterization of all associated viruses in a sample. However, metagenomic analysis is not a feasible technique for diagnostic laboratories that test thousands of field samples annually because of the per-sample cost, the labor involved in sample preparation, and the sophisticated equipment required. Culture is not ideal, since its sensitivity is dependent on the culture system(s) used and the growth characteristics of the virus(es) present in a sample. Nonetheless, primary culture still allows for the detection and isolation of some novel or unexpected viruses that would otherwise remain unknown.

## Figures and Tables

**Figure 1 viruses-11-00927-f001:**
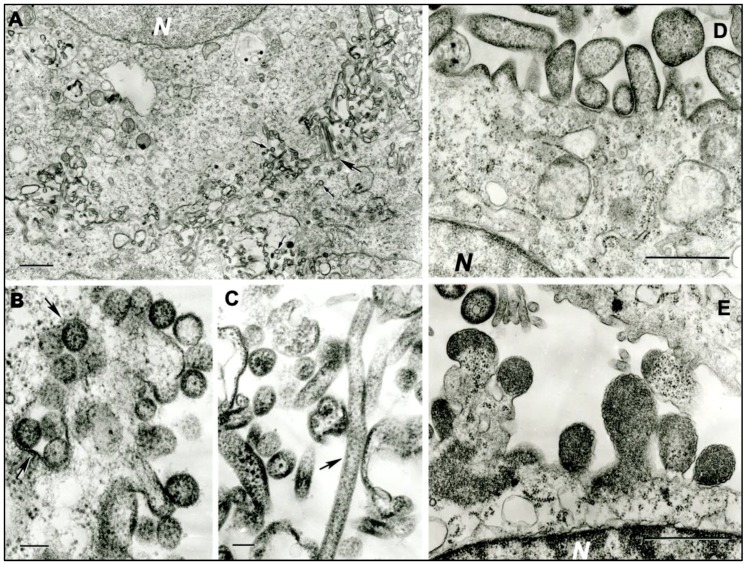
Ultrastructure of MCRV (DO 159) in infected C6/36 cells, and SJCV (DO 200) in infected Vero E6 cells. (**a**) Multiple virus particles of MCRV forming at the cell surfaces of adjacent cells. N- cell nucleus. The large arrow shows two very long filamentous virions; small arrows show spherical virions. Bar = 1 µm. (**b**) Spherical virions of MCRV 130–150 nm in diameter at the cell surface. Arrows show cross-sections of the invaginations of the cell surface containing virions. Bar = 100 nm. (**c**) Virions of MCRV in extracellular space, a very long one (arrow) and pleomorphic. Bar = 100 nm. (**d**,**e**) Virions of SJCV forming at the surface of infected Vero E6 cells. *N* = fragments of cell nuclei. Bar = 1 µm.

**Figure 2 viruses-11-00927-f002:**
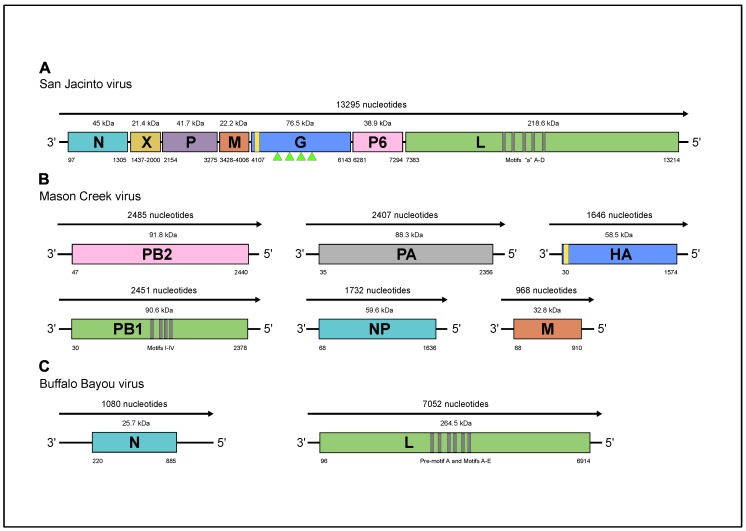
Schematic representation of the genome organizations (**a**) SJCV; (**b**) MCV; and (**c**) BBAV. Green triangles indicate the location of predicted glycosylation sites.

**Figure 3 viruses-11-00927-f003:**
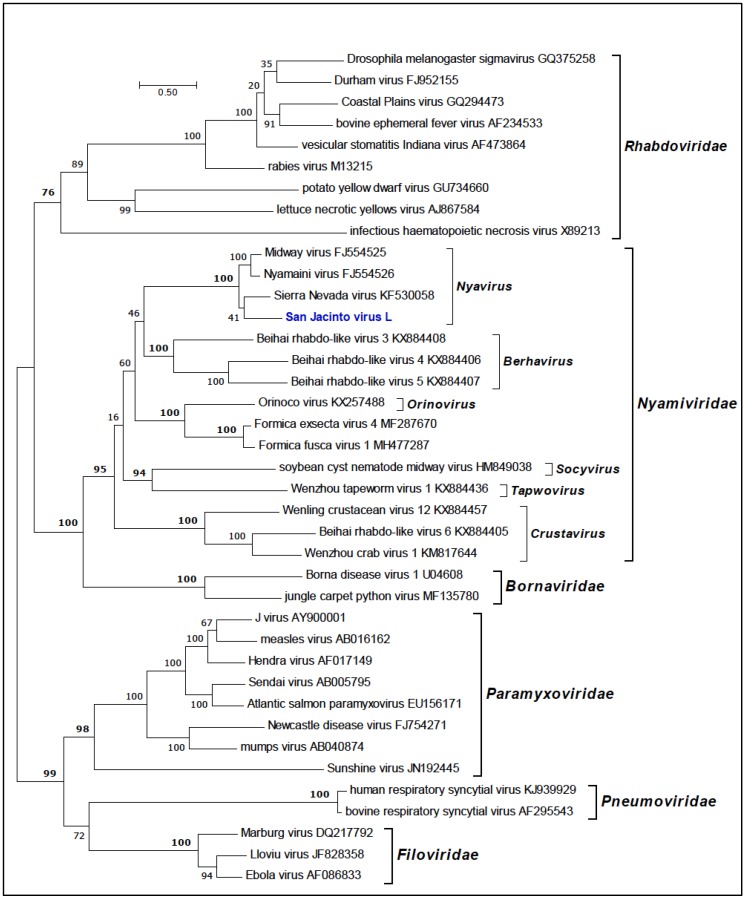
Maximum likelihood phylogenetic tree of the RNA-dependent RNA polymerase (L protein) amino acid sequences for representative members of the order *Mononegavirales* with emphasis on the family *Nyamiviridae*. Branch lengths are proportional to the number of nucleotide substitutions per site. Values for bootstrap support proportion (BSP) were determined from 100 bootstrap iterations. BSP values showing strong support for each of the key nodes are shown in bold. The position in the tree of SJCV is shown in blue. The Genbank accession numbers of all sequences are as shown.

**Figure 4 viruses-11-00927-f004:**
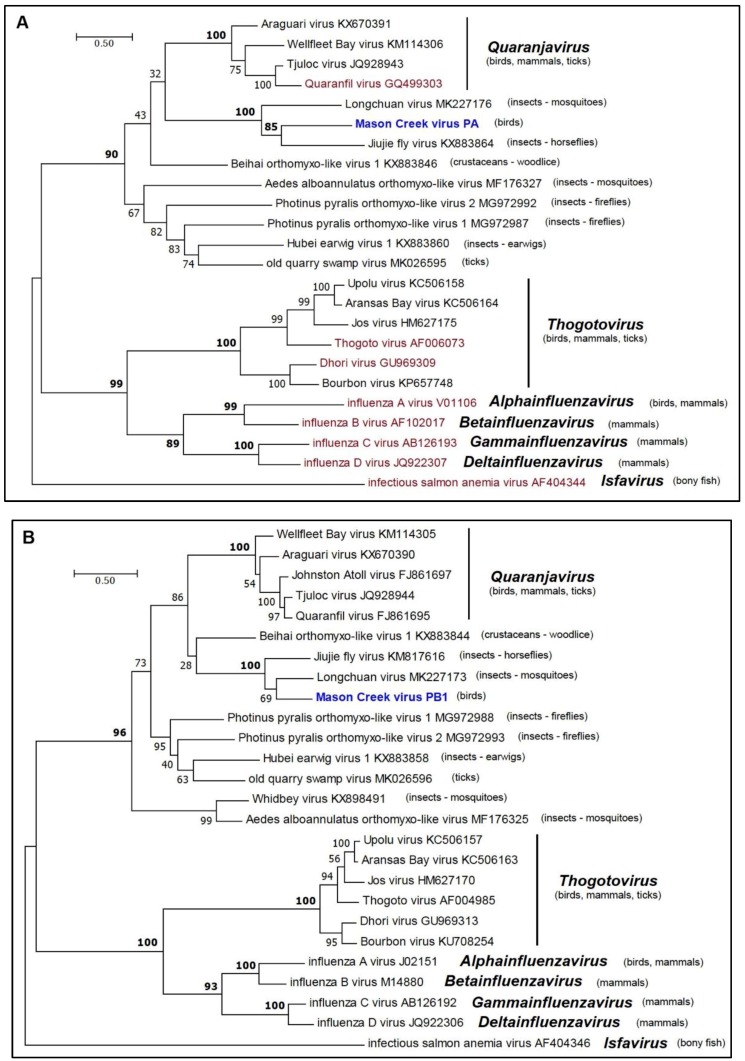
Maximum likelihood phylogenetic trees for amino acid sequences of representative members of the family *Orthomyxoviridae*. (**a**) PA proteins; (**b**) PB1 proteins; (**c**) PB2 proteins; and (**d**) NP proteins. Branch lengths are proportional to the number of nucleotide substitutions per site. Values for bootstrap support proportion (BSP) were determined from 100 bootstrap iterations. BSP values showing strong support for each of the key nodes are shown in bold. In each tree, viruses formally assigned to species are shown in magenta and the position of MCV is shown in blue. The Genbank accession numbers of all sequences are as shown.

**Figure 5 viruses-11-00927-f005:**
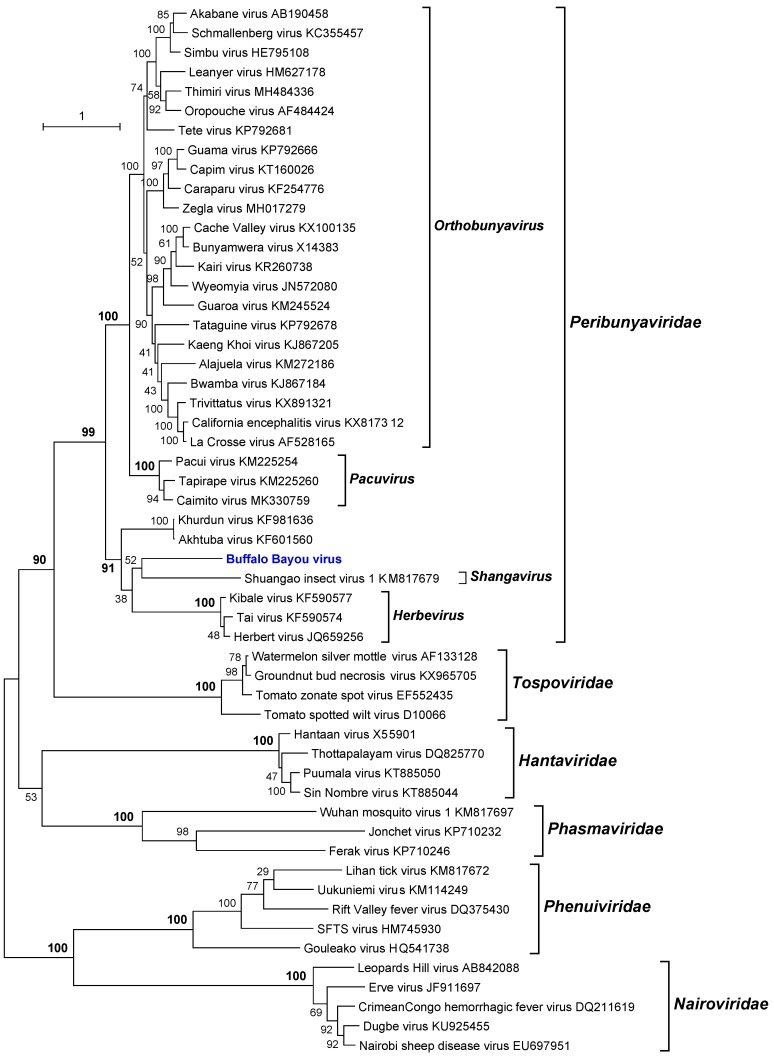
Maximum likelihood phylogenetic tree of the RNA-dependent RNA polymerase (L protein) amino acid sequences for representative members of the order *Bunyavirales* with emphasis on the family *Peribunyaviridae*. Branch lengths are proportional to the number of nucleotide substitutions per site. Values for bootstrap support proportion (BSP) were determined from 100 bootstrap iterations. BSP values showing strong support for each of the key nodes are shown in bold. The position in the tree of BBAV is shown in blue. The Genbank accession numbers of all sequences are as shown.

**Table 1 viruses-11-00927-t001:** Summary of tree novel viruses isolated from brains of dead birds in Harris County, TX.

Field Number	Proposed Virus Name	Abbreviation	Avian Source, Date Collected	Proposed Classification (Genus; Family)
DO-200	San Jacinto virus	SJCV	*Sturnus vulgaris*, 20 June 2013	*Nyavirus*; *Nyamiviridae*
DO-159	Buffalo Bayou virus	BBAV	*Cyanocitta cristata**, 27 July 2011	*(uncertain)*; *Peribunyaviridae*
DO-159	Mason Creek virus	MCRV	*Cyanocitta cristata**, 27 July 2011	*Quaranjavirus*; *Orthomyxoviridae*

* Bird brain decomposing with fly larvae.

**Table 2 viruses-11-00927-t002:** Complement fixation tests of San Jacinto virus (SJCV) with Nyamanini (NYMV), Midway (MIDWV), Sierra Nevada (SNVV), and Newcastle disease (NDV).

Antigen	Antibody				
	NYMV	MIDWV	SNVV	NDV	SJCV
**NYMV**	512/128*	0	0	0	0
**MIDWV**	0	1024/≥512	0	0	0
**SNVV**	0	0	64/>4	0	0
**NDV**	0	0	0	≥64/≥4	0
**SJCV**	0	0	0	0	128/≥4

* Reciprocal of highest antiserum dilution/highest antigen dilution 0 =< 4/4.

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
