# Peer review of "Characterization of Three Novel Viruses from the Families Nyamiviridae, Orthomyxoviridae, and Peribunyaviridae, Isolated from Dead Birds Collected during West Nile Virus Surveillance in Harris County, Texas"

_viruses, 2019, doi:10.3390/v11100927_

Round 1

Reviewer 1 Report

The study submitted by P.T. Walker and collaborators identified by EM and NGS three novel viruses from wild bird specimens recently collected during arbovirus surveillance in Texas, USA. Full-genome sequencing allowed characterizing virus genome organization and inferring phylogenetic relatedness to close viruses. This study is well designed and brings additional and original information on the avifauna virosphere, even though it does not address the clinical impact of the three viral infections. 

Minor modifications are proposed : 

-line 76, Qiagen

-line 118 : were raw data (reads) obtained from HiSeq also deposited on public databases?

-line 123 : "conserved motifs were annotated"

-Figure 1 legend ; please use MCRV as elsewhere in the text

-line 183, "search of the contigs against"

-Figure 2 legend should be completed (triangles in the G protein of SJCV)

-line 225 : "PA encoding a 773 aa" and "a putative protein of 782 aa"

-line 230: WFBV not presented earlier in the text; "which envodes a putative"

-line 255, explain why CFTs were performed. 

-lines 265 and 267 : "Nyamiviridae"

-please enlarge figure 4

-line 295 "NP"

-line 310 : BBAV could also cluster into shangaviruses, more genetic data are needed to classify BBAV

-The authors do not discuss their results in the light of previously published works. Appropriately discuss the findings (identification of close viruses in birds or other taxons, possibility to predict the virulence of the identified viruses for birds,...)

-line 343 : "conserved between the nyavirus L proteins"

-lines 350 and 351 should be BBAV instead of MCRV

Author Response

The study submitted by P.T. Walker and collaborators identified by EM and NGS three novel viruses from wild bird specimens recently collected during arbovirus surveillance in Texas, USA. Full-genome sequencing allowed characterizing virus genome organization and inferring phylogenetic relatedness to close viruses. This study is well designed and brings additional and original information on the avifauna virosphere, even though it does not address the clinical impact of the three viral infections. 

Minor modifications are proposed: 

-line 76, Qiagen

Response: corrected as suggested

-line 118: were raw data (reads) obtained from HiSeq also deposited on public databases?

Response: The raw data have not been deposited to public databases. However, they are available upon request to the corresponding author. We have now included the sentence: “Raw sequencing data are available upon request to the corresponding author”

-line 123: "conserved motifs were annotated"

Response: corrected as suggested

-Figure 1 legend; please use MCRV as elsewhere in the text

Response: corrected as suggested

-line 183, "search of the contigs against"

Response: it is unclear as to what the reviewer suggests here. The authors believe that the sentence structure as is accurately depicts our methodology.

-Figure 2 legend should be completed (triangles in the G protein of SJCV)

Response: we thank the reviewer for catching our omission. We have now added the following text on lines 197-198 “which contains four predicted glycosylation sites (triangles Figure 2a)”

-line 225: "PA encoding a 773 aa" and "a putative protein of 782 aa"

Response: corrected as suggested

-line 230: WFBV not presented earlier in the text; "which encodes a putative"

Response: corrected as suggested

-line 255, explain why CFTs were performed. 

Response: updated as requested

-lines 265 and 267: "Nyamiviridae"

Response: corrected as suggested

-please enlarge figure 4

Response: Amended as suggested

-line 295 "NP"

Response: corrected as suggested

-line 310: BBAV could also cluster into shangaviruses, more genetic data are needed to classify BBAV

Response: corrected as suggested

-The authors do not discuss their results in the light of previously published works. Appropriately discuss the findings (identification of close viruses in birds or other taxons, possibility to predict the virulence of the identified viruses for birds,...)

Response: Predicting virulence in birds based on relationships (genetic or otherwise) to related viruses is pure speculation. However, we would like to bring to the reviewer’s attention that related nyaviruses to SJCV, like Midway (MIDWV), Sierra Nevada (SNVV) and Nyamanini viruses, have all been isolated from birds and/or bird ticks; but none of them have been isolated from sick or dead birds.

-line 343: "conserved between the nyavirus L proteins"

Response: corrected as suggested

-lines 350 and 351 should be BBAV instead of MCRV

Response: corrected as suggested

Reviewer 2 Report

The manuscript presents an extensive and detailed research which is well written. Nevertheless there is one issue to be addressed. The isolate DO200 containing San Jacinto virus (SJV) is isolated from an intact bird brain, produces CPE in mammalian Vero cells but no CPE in insect C6/36 cells and produced illness in inoculated newborn mice. Beside that SJV is related to other bird viruses and all these strongly suggest that SJV infects birds. On the other hand, the DO 159 isolate containing Mason Creek virus (MCV) and Buffalo Bayou virus (BBV) is obtained from a brain of partially decomposed bird carcass with unidentified maggots observed in the brain. The DO 159 isolate does not produce CPE in Vero cells, but produces CPE in C6/36 cell. It did not produce illness in inoculated newborn mice. Both viruses detected in this sample, MCV and BBV, are not closely related to avian viruses, but rather to invertebrate, particularly insect viruses. Until further advances in knowledge of MCV and BBV will be available there is a room for assumption that MCV and BBV might originate from the maggots and not from the bird itself. This should be addressed in the manuscript. In that way the article title perhaps should refer to bird carcasses instead of birds.

There are also few minor language errors to be addressed e.g. revising the sentence in the rows 122-123; changing “phylo genetic” into “phylogenetic” in the row 300;  capitalization of “sierra nevada” in the row 423 as it is originally written in the article referenced etc.

Author Response

The manuscript presents an extensive and detailed research which is well written. Nevertheless, there is one issue to be addressed. The isolate DO200 containing San Jacinto virus (SJV) is isolated from an intact bird brain, produces CPE in mammalian Vero cells but no CPE in insect C6/36 cells and produced illness in inoculated newborn mice. Beside that SJV is related to other bird viruses and all these strongly suggest that SJV infects birds. On the other hand, the DO 159 isolate containing Mason Creek virus (MCV) and Buffalo Bayou virus (BBV) is obtained from a brain of partially decomposed bird carcass with unidentified maggots observed in the brain. The DO 159 isolate does not produce CPE in Vero cells, but produces CPE in C6/36 cell. It did not produce illness in inoculated newborn mice. Both viruses detected in this sample, MCV and BBV, are not closely related to avian viruses, but rather to invertebrate, particularly insect viruses. Until further advances in knowledge of MCV and BBV will be available there is a room for assumption that MCV and BBV might originate from the maggots and not from the bird itself. This should be addressed in the manuscript. In that way the article title perhaps should refer to bird carcasses instead of birds.

Response: We thank the reviewer for raising this important observation. Mason creek and Buffalo bayou viruses may well be insect viruses, based on their phenotypic properties and phylogenetic relationships. We have now included text to articulate it within the manuscript. We have also modified the title of the manuscript for accuracy. 

There are also few minor language errors to be addressed e.g. revising the sentence in the rows 122-123; changing “phylo genetic” into “phylogenetic” in the row 300; capitalization of “sierra nevada” in the row 423 as it is originally written in the article referenced etc.

Response: corrected as suggested

Reviewer 3 Report

Walker et al describe the isolation of viruses from two deceased birds in Harris county Texas from 2002 to 2014. The authors isolate, identify and briefly describe three novel viruses within the families Nyamaviridae, Orthomyxoviridae and Peribunyaviridae.

Overall the study is well conducted with the material and data available. The conclusions based upon available evidence are for the most part supported by the data. Specific concerns are provided below.

Readability could be improved by providing a table with the sample name, the proposed viral name and family isolated from the sample. The introduction should include an additional sentence or two describing the surveillance program to bring it in line with the discussion of the merits of tissue culture procedures used in surveillance. How did the authors account for the possibility that MCV or BBAV was derived from the fly larvae, and why CPE was observed in C6/36 cells but not vero cells. Based upon the phylogenetic analysis the authors note that MCRV clustered with Jiujie fly virus, which is known to infect flies. Did the sequence analysis indicate the presence of DNA viruses? How do the authors know the virions observed from DO159 are from MCV and not BBAV? The spherical virions in fig 1b resemble HEBV virions1. The label for the EM panels or the nucleus should be changed. The font indicating the nucleus is very similar. Figure legend 2 needs to be improved. For example. What does the green arrows indicated in Fig 2a? An indication if surveillance procedures included diagnostics approaches that would identify MCRV, BBAV or SJCV in other bird samples from 2002 to 2014. The authors should expand the discussion to include information how the phylogenic analysis contributes to the understanding of the Nyamaviridae, Orthomyxoviridae and Peribunyaviridae virus families. Line 122/3 needs to be rewritten.

1 Marklewitz M, Zirkel F, Rwego IB, et al. Discovery of a unique novel clade of mosquito-associated bunyaviruses. J Virol. 2013;87(23):12850–12865. doi:10.1128/JVI.01862-13

Author Response

Walker et al describe the isolation of viruses from two deceased birds in Harris county Texas from 2002 to 2014. The authors isolate, identify and briefly describe three novel viruses within the families Nyamaviridae, Orthomyxoviridae and Peribunyaviridae.

Overall the study is well conducted with the material and data available. The conclusions based upon available evidence are for the most part supported by the data. Specific concerns are provided below.

Readability could be improved by providing a table with the sample name, the proposed viral name and family isolated from the sample.

Response: We have now included a table with the requested information

The introduction should include an additional sentence or two describing the surveillance program to bring it in line with the discussion of the merits of tissue culture procedures used in surveillance.

Response:  We have to disagree with the reviewer here as we briefly describe the surveilance program and give 3 references that describe it more fully. The comment about the additional value of cell culture was not a hypothesis or objective of the surveillance program, it was just an observation we made at the end

How did the authors account for the possibility that MCV or BBAV was derived from the fly larvae, and why CPE was observed in C6/36 cells but not vero cells. Based upon the phylogenetic analysis the authors note that MCRV clustered with Jiujie fly virus, which is known to infect flies.

Response: We have now included text in the manuscript to address this issue

Did the sequence analysis indicate the presence of DNA viruses?

Response: We did not detect any DNA viruses, and it should be noted that our sample and library prep methods are biased towards RNA

How do the authors know the virions observed from DO159 are from MCV and not BBAV?

The spherical virions in fig 1b resemble HEBV virions1.

Response: Members of the Peribunyaviridae have a smaller diameter (<100 nm), whereas members of the Orthomyxoviridae have a larger diameter.

The label for the EM panels or the nucleus should be changed. The font indicating the nucleus is very similar.

Response: corrected as suggested

Figure legend 2 needs to be improved. For example. What does the green arrows indicated in Fig 2a?

Response: We have now included text to indicate that green arrows represent predicted glycosylation sites

An indication if surveillance procedures included diagnostics approaches that would identify MCRV, BBAV or SJCV in other bird samples from 2002 to 2014. The authors should expand the discussion to include information how the phylogenic analysis contributes to the understanding of the Nyamaviridae, Orthomyxoviridae and Peribunyaviridae virus families.

Response: Prior to 2012, CF tests (or other serologic tests, such as HI) were routinely performed in our program to identify viruses based on their serological properties and confirmed by TEM. Since 2013 we also included NGS to obtain their sequence and assign them taxonomically based on their genomic sequences. These were the only viruses other than West Nile that we identified in birds. 

Line 122/3 needs to be rewritten.

Response: corrected as suggested

Marklewitz M, Zirkel F, Rwego IB, et al. Discovery of a unique novel clade of mosquito-associated bunyaviruses. J Virol. 2013;87(23):12850–12865. doi:10.1128/JVI.01862-13